# Genomic Characterisation of a Multiple Drug Resistant IncHI2 ST4 Plasmid in *Escherichia coli* ST744 in Australia

**DOI:** 10.3390/microorganisms8060896

**Published:** 2020-06-14

**Authors:** Tiziana Zingali, Toni A. Chapman, John Webster, Piklu Roy Chowdhury, Steven P. Djordjevic

**Affiliations:** 1The ithree Institute, University of Technology Sydney, City Campus, Ultimo, NSW 2007, Australia; tiziana.zingali@student.uts.edu.au (T.Z.); Piklu.Bhattacharya@uts.edu.au (P.R.C.); 2Australian Centre for Genomic Epidemiological Microbiology, University of Technology Sydney, P.O. Box 123, Broadway, NSW 2007, Australia; 3NSW Department of Primary Industries, Elizabeth MacArthur Agricultural Institute, Menangle, NSW 2568, Australia; toni.chapman@dpi.nsw.gov.au (T.A.C.); john.webster@dpi.nsw.gov.au (J.W.)

**Keywords:** IncHI2, plasmid, porcine, *mefB*, complex resistance locus, multiple drug resistance

## Abstract

Antibiotic resistance genes (ARGs) including those from the *bla*_CTX-M_ family and *mcr-1* that encode resistance to extended spectrum β–lactams and colistin, respectively, have been linked with IncHI2 plasmids isolated from swine production facilities globally but not in IncHI2 plasmids from Australia. Here we describe the first complete sequence of a multiple drug resistance Australian IncHI2-ST4 plasmid, pTZ41_1P, from a commensal *E. coli* from a healthy piglet. pTZ41_1P carries genes conferring resistance to heavy-metals (copper, silver, tellurium and arsenic), β-lactams, aminoglycosides and sulphonamides. The ARGs reside within a complex resistance locus (CRL) that shows considerable sequence identity to a CRL in pSDE_SvHI2, an IncHI2:ST3 plasmid from an enterotoxigenic *E. coli* with serotype O157:H19 of porcine origin that caused substantial losses to swine production operations in Australia in 2007. pTZ41_1P is closely related to IncHI2 plasmids found in *E. coli* and *Salmonella enterica* from porcine, avian and human sources in Europe and China but it does not carry genes encoding resistance to clinically-important antibiotics. We identified regions of IncHI2 plasmids that contribute to the genetic plasticity of this group of plasmids and highlight how they may readily acquire new resistance gene cargo. Genomic surveillance should be improved to monitor IncHI2 plasmids.

## 1. Introduction

Vast amounts of antimicrobial residues, heavy-metals and multi-drug resistant (MDR) bacteria are released into the environment each year [1]. Whole genome sequencing (WGS) is now the leading technology to perform genomic epidemiology and garner deep insights into pathogen evolution and drug resistance gene flow [2,3,4]. *Escherichia coli* is likely to encounter and adapt to antimicrobial selection pressures because it is widespread in the gut of mammals and to a lesser extent reptile, diverse commercial and wild avian species as well as in the environment and food [3]. As such, *E. coli* is a logical target for genomic surveillance studies seeking to understand antimicrobial resistance from a One Health perspective. 

Combating *E. coli* in veterinary settings is a growing concern as last-line antimicrobials such as colistin and third-generation cephalosporins are used to treat infectious diseases in intensively reared production animals in many regions of the world, particularly in Southeast Asia [5]. Australian food-animal production systems have been judicious in the use of last-resort antibiotics to treat infections in humans. Fluoroquinolones, colistin and fourth-generation cephalosporins have never been approved for use in Australian food animals, although usage of third-generation cephalosporins such as ceftiofur is allowed for off-label use in pigs [6]. Nonetheless, phenotypic and genotypic resistance to many first-generation antimicrobials is now widespread in commensal faecal *E. coli* populations in healthy Australian swine [7,8]. In intensive pig farming systems, antimicrobials are used to prevent the spread of diseases and animal loss [9,10]. Sows are an important reservoir of bacteria that carry antibiotic and heavy metal resistance genes which are readily transmitted to offspring [9,11,12]. Animal diet, production cycle stage, and poor farm biosecurity routines are also likely to contribute to transmission and propagation of MDR *E. coli* on farm and in the environment [13,14]. 

*E. coli* ST744 is increasingly resistant to multiple antimicrobials and is recognised as a pathogen in animal production systems, dogs and in wildlife [15,16,17]. In Europe, *E. coli* ST744 is known to harbour genes conferring resistance to colistin and carbapenems on different Inc-type plasmids [18]. In Australia, *E. coli* ST744 sourced from pigs has been reported with low frequency exhibiting resistance to extended-spectrum cephalosporins and fluoroquinolones [7]. 

Plasmids play a central role in the capture of antimicrobial resistance and virulence genes. In swine, numerous plasmid families have been described, but those belonging to the IncF and IncHI2 incompatibility groups are prominent in their role as purveyors of virulence and antimicrobial resistance genes [8,19]. In *Enterobacteriaceae*, IncHI2 plasmids were reported as the fifth most common plasmid family [20] and predominantly belonged to ST3 and ST4 according to the IncHI2 dichotomous pMLST system [21]. They are usually large plasmids (>250 kb), with optimum conjugation temperatures below 37 °C [20,22], which suggests that these plasmids may conjugate under a wide range of temperatures and in the environments like faecal holding ponds, agricultural lands and waterways. IncHI2 plasmids are characterised by the presence of transposons that carry multiple drug and heavy-metal resistance genes [19,23,24,25]. Antimicrobial resistance genes (ARGs) are most frequently part of class 1 integrons associated with various transposons, mostly belonging to the Tn*3* family. IS*26* plays an important role in the capture and assembly of ARGs and complex resistance locus (CRL) within transposons [19,23,25] and in shaping the structure of class 1 integrons [8,25,26,27,28]. 

IncHI2 plasmids circulating within MDR Australian isolates have been reported to carry genes conferring resistance to β- lactams, aminoglycosides, trimethoprim, sulphonamides, as well as heavy metals, such as copper, silver and arsenic [19,23,24,25,29]. IncHI2 is one of the major plasmid groups that carries *mcr-1* gene variants conferring resistance to colistin, a last-line antimicrobial used in the treatment of MDR bacterial infections. 

Here we report the complete sequence of pTZ41_1P_HI2, the first MDR IncHI2 ST4 plasmid characterised from a porcine commensal in Australia. pTZ41_1P_HI2 was isolated from an *E. coli* collected from the faeces of a healthy four-week old piglet in 2017. We present the phylogeny of pTZ41_1P_HI2 and other IncHI2 ST4 plasmid sequences available in public repositories. We also present a detailed comparison of the drug resistance locus in pTZ41_1P with that found in closely related plasmid pSDE-SvHI2, an IncHI2:ST3 plasmid isolated from a pathogenic porcine ETEC/ExPEC in Australia in 2007 [19,30]. This analysis will provide insight into how MDR IncHI2 plasmids that have been circulating in commercial pig production environments in Australia over a 10-year period have evolved.

## 2. Materials and Methods

### 2.1. Plasmid pTZ41_1P_HI2

pTZ41_1P_HI2 was identified from a commensal phylogroup A *E. coli* (strain TZ41_1P) belonging to ST744 and serotype O89:H25. The strain was sourced from faecal material of a four-week old piglet, and part of a collection of commensal *E. coli* that carry the class 1 integrase *intI1^+^*. The cohort of pigs was sourced from an Australian commercial breeding operation in January 2017. Pigs were not previously treated with antimicrobials and weaned on the day before the initial faecal sampling.

### 2.2. Sequencing and Assembly of Plasmid pTZ41_1P_HI2

The complete sequence of pTZ41_1P_HI2 was assembled using the Unicycler v0.4.6 [31] hybrid assembler with raw reads from Illumina and Nanopore sequencing runs generating a completely closed plasmid sequence. For illumina short read sequencing, genomic DNA from *Escherichia coli* TZ41_1P was isolated using a DNeasy Blood and Tissue Kit (Qiagen) following the manufacturer’s protocol. DNA quantification and library preparation were performed as described previously [8]. DNA for Nanopore sequencing was isolated with a Genomic tip 20/G kit (Qiagen) following the manufacturer’s protocol. MinION sequencing was performed at EMAI (Menangle, NSW) with barcoded sequences using Oxford Nanopore’s PCR-free ligation sequencing kit (SQK-LSK109) and Native Barcoding expansion kit (EXP-NBD104) and loaded onto a R9.4.1 flow cell. Libraries were prepared according to manufacturer’s protocol with the DNA fragmentation step being skipped and a long fragment selection performed prior to library loading. Flow cell QC showed 1614 active pores and sequencing with MinKnow was run for 48 hrs. Base calling through MinKnow was performed live and resulting fastq files were then used for assembly [31]. The DNA sequence of pTZ41_1P has been deposited in GenBank (NCBI) under Accession number MT604108 

### 2.3. Gene Screening, Plasmid Typing and Alignment Analysis

Antimicrobial resistance genes and *E. coli* sequence type (ST) were determined through the various portals made available via the Centre of Genomic Epidemiology website (http://www.genomicepidemiology.org/). Plasmid incompatibility groupings were determined using PlasmidFinder and plasmid sequence type following the IncHI2 pDLST scheme. Preliminary plasmid annotations were generated using RASTtk (https://rast.nmpdr.org/) and Galileo^TM^ AMR (formerly MARA) database (galileoamr.arcbio.com/mara/). Single gene annotations were manually curated using BLASTn (https://blast.ncbi.nlm.nih.gov/Blast.cgi). Insertion sequences were identified and manually annotated using ISfinder (https://www-is.biotoul.fr/index.php). Snapgene v4.1.9 was used to create the final annotation file.

IncHI2 ST4 plasmid sequences were retrieved from GenBank (Appendix A) using BLASTn. The search parameters were ≥95% identity over 100% input query length using smr0018 (allele n.4) and smr0199 (allele n.2) as the search query. Allele 4 of smr0018 and allele 6 of smr0199 form the basis of subtyping IncHI2 in the pDLST scheme [21]. Additionally, all the plasmid sequences were submitted to (https://pubmlst.org/plasmid/) for confirmation of the IncHI2 double-locus sequence type (pDLST). The phylogeny of core plasmid genes was investigated using parSNP [32] with the -x flag to filter recombination events, and visualised in FigTree v1.4.3 (http://tree.bio.ed.ac.uk/software/figtree/). Genomic alignments and comparison figures were generated using BRIG v0.95 [33] and Easyfig v2.2.2 [34]. For comparative analyses with pSDE-SvHI2 sequence, the annotation file was downloaded from GenBank (accession numbers MH287084).

## 3. Results

### 3.1. Gene Content of pTZ41_1P_HI2

Plasmid pTZ41_1P_HI2 was 269,099 bp in length and typed as an IncHI2 sequence type 4 (ST4) plasmid (Figure 1, outer ring). The backbone of the plasmid comprised transfer (*trh* genes), replication (*repHI2*) and partitioning (*par*) gene homologues characteristic of IncHI2 plasmids. The plasmid also carried tellurium resistance genes (*terZABCDEF*), commonly associated with IncHI2 plasmids, in addition to *terY1*, *terY2* and *terW.* pTZ41_1P_HI2 also carries the operons encoding *sil* and *pco,* that confer resistance to copper and silver, in association with a Tn*7*-like transposon. Additionally, genes conferring arsenic resistance (*arsCBRH*) were found in pTZ41_1P_HI2, 11,613 bp away from the *sil/pco* cluster. The arsenic resistance operon was previously reported in R478, the prototype IncHI2 plasmid [35]. There is no evidence of association of transposition genes with the *ars* operon in pTZ41_1P_HI2. An alignment of phylogenetically related IncHI2:ST4 plasmid sequences highlighted the absence of the *ars* operon (Figure 1, outer ring). 

A 42,954 bp complex resistance locus (CRL) was identified in pTZ41_1P_HI2. The CRL was housed within a truncated Tn*7*-like transposon (Figure 2) which contained a hybrid Tn*1721*/Tn*21* transposon inserted in *tnsA*. The CRL comprised genes capable of conferring resistance to trimethoprim (*dfrA12*), chloramphenicol (*cmlA1*), ampicillin (*bla*_TEM-1_) aminoglycoside antibiotics streptomycin (*aadA1*, *strA* and *strB*) and spectinomycin (*aadA1*) and sulfonamides (*sul2*, *sul3*). Insertion of the hybrid Tn*1721*/Tn*21* was evidenced by 6 bp DR abutting IRRII and IRL of Tn*1721* (Figure 1). Additional characteristic features of the CRL included a *sul3*-associated class 1 integron and a variant of Tn*6029* (Figure 2 and Figure 3). 

The class 1 integron was inserted in the Tn*21* backbone at a site identical to In*2* integrons. A 5 bp direct repeat was identified that is characteristic of In*2* integrons (Figure 2). The presence of a 7 bp sequence (GTTAGTC) in between the *qacH* cassette and *sul3* gene was previously identified in pCERC3 (*53*), also highlights an association of the *sul3*-integron with Tn*21* (*54*). *dfrA12-orfF-aadA2-cmlA1-aadA1-qacH* comprised the cassette array in the integron followed by a putative *tnp440* transposase and a *sul3-orfB-orfA-mefB* module (Figure 2). Previous research described *sul3*-integrons on IncHI2 ST3 plasmids from Australian porcine *E. coli* [19]. In addition to the *sul3*-integron, the hybrid Tn*1721*/Tn*21* transposon comprised an IS*26*-flanked Tn*6029-like* transposon (Figure 3) similar to those previously reported in diverse lineages of pathogenic *E. coli* sourced from humans and livestock in Australia [27,36,37] and overseas [38,39]. 

Tn*6029* is a compound transposon made up of two modules separated by three copies of IS*26*. The streptomycin resistance module consisting of *sul2*, *strA* and *strB* genes is thought to have originated from RSF1010 and is mobilised by IS*26* [40]. A second module with the *bla*_TEM-1b_ gene (encodes resistance to ampicillin) is derived from a Tn*2* transposon [41]. The IS*26* adjacent to the *strB* gene in pTZ41_1P_HI2 is located 291 nucleotides into *tnpA* and the orientation of the IS*26* is inverted when compared to other transposons of the Tn*6029* family [38]. The Tn*6029*-family of transposons are often described in association with specific IS*26*-mediated insertions into or abutting class 1 integrons. These transposons are known to be carried by different plasmid types and within genomic islands, often in association with Tn*21* [27,37,38,42]. A BLASTn analysis of the region spanning the 5′-CS of the class 1 integron and the *mer*-module of Tn*21* in pTZ41_1P_HI2 showed that the association between the *sul3*- integron and a Tn*6029-like* transposon has not been reported to date.

### 3.2. Phylogeny and Comparative Genomic Analysis of IncHI2 ST4 Plasmids in Genbank Database

A BLASTn search using the IncHI2 ST4 pDLST-specific alleles as the query identified 14 ST4 plasmids sequences (Appendix A, BLASTn search April 2019). In-silico PCR of the five target genes used for pDLST subtyping [20] showed an identical profile for all 14 entries including pTZ41_1P_HI2 (Appendix A). However our SNP-based phylogeny analysis using pSSE-ATCC-43845, the oldest plasmid sequence (1941) reported in GenBank [1] revealed that only 6/14 plasmids shared ancestral relationships (Figure 4). The inference was based on alignment of all 6 plasmids over 49% of the length of reference plasmid with the recombination filtering mode turned on. pTZ41_1P_HI2 appeared to be a distinct plasmid lineage when compared to the rest of the plasmids sourced from Europe between 2008 and 2016. Although a limited number of complete plasmid sequences were available, our results indicate that the Australian ST4 lineage may have evolved distinctly. Nevertheless, we recommend that further sequencing of IncHI2 ST4 plasmids is needed to comprehend their distribution and phylogenetic associations in Australia.

A pair-wise BLAST analysis of IncHI2 ST4 plasmid sequences showed high sequence identity between pTZ41_1P_HI2 and pRDB9 (Figure 1, inner circles). We found that pTZ41_1P_HI2 was 99.7% identical to pRDB9 over 91% of their sequence length, with sequence differences largely confined to laterally acquired regions associated with resistance determinants. Plasmid pRDB9 was sourced from a *mcr-1* positive ST10 *E. coli* RDB9 in 2015 from a slaughtered pig in Switzerland [43]. With the exception of *mcr-1*, most of the extensive repertoire of ARGs carried on this resistance resided in a 60 kDa CRL included two class 1 integrons, Tn*6029*, Tn*5393*, and Tn*1721* or variants thereof. The plasmid also contained heavy metal resistance loci [43]. The *sul3*- integron in pRDB9 was flanked by IS*26*, and contained the ∆*mefB*_260_ signature previously reported in Australian commensal *E. coli* of porcine origin [8].

### 3.3. Comparative Analysis of sul3 Integrons in pTZ41_1P_HI2 and pSDE-SvHI2

We conducted an analysis of the ARG content between two Australian MDR *sul3*-carrying IncHI2 plasmids of porcine origin, pTZ41_1P and pSDE-SvHI2. pSDE-SvHI2 is an IncHI2:ST3 plasmid sourced from *E. coli* O157 strain SvETEC from a diseased piglet in 2007 in New South Wales, Australia (Figure 3) [19,30]. O157 SvETEC is ST4245 (ST23 clonal complex) that is representative of an *E. coli* clone that caused significant economic losses and multiple fatalities in affected piggeries in the mid-2000s in Australia [30]. In contrast to pTZ41_1P_HI2, the *sul3*-associated class 1 integron in pSDE-SvHI2 appeared to be mobilised by IS*26*, but the mechanism of insertion into the Tn*7-like* transposon could not be clearly defined [19]. Additionally, the *sul3*- integron in pSDE-SvHI2 carried IS*26*-mediated deletions, such as the truncation of the integrase gene *intI1* (Δ*intI1_705_*) and the macrolide efflux gene *mefB* (Δ*mefB_111_*), which we described as genomic signatures characteristic of the CRL in pSDE-SvHI2. Finally, the *sul3*-associated class 1 integron in pSDE-SvHI2 harboured a different set of gene cassettes in the cassette array compared to pTZ41_1P_HI2 and carried a variant of Tn*5393* absent in pTZ41_1P_HI2. A second class 1 integron was also identified in a Tn*1721*/Tn*21* hybrid transposon in pSDE-SvHI2.

## 4. Discussion

IncHI2 plasmids are known to be important purveyors of clinically important ARGs encoding resistance to carbapenems [29], colistin [44], and quinolones [45]. Recently MDR IncHI2 ST4 plasmids ranging in size from 235,260 to 356,700bp that carry *mcr-1* sourced from *E. coli* with ST10 from pigs with post-weaning diarrhoea, and *E. coli* ST10 and ST351 from white storks were described in Spain [46]. ST10 and other clonal complex 10 *E. coli* are frequently sourced from the faeces of pigs, fresh produce, intensively reared food-production animals, and diverse environmental niche [46,47,48] and often carry MDR IncHI2 plasmids. Colistin resistance *mcr* genes have not been reported from Australian porcine *E. coli* isolates. pTZ41_1P_HI2 is an ST4 plasmid which carries ARGs conferring resistance to various first-line antimicrobial agents. The genes are found in a CRL and their assembly has in part been mediated by the action of IS*26,* [49] an insertion element that has been linked to their capture, assembly and mobilisation of most ARGs, [27] including those encoding resistance to critically-important antibiotics [25]. In addition IS*26* plays a role in the formation of hybrid plasmids [50] and instances where virulence and resistance plasmids have fused are known [51]. As such, the presence of IS*26* in pTZ41_1P_HI2 represents a potential hotspot for the introduction of new AGR and virulence gene cargo.

The post-weaned piglet carrying *E. coli* TZ41_1P did not directly encounter an antimicrobial treatment, nor did it come into contact with grain-based feed formulated with commonly used heavy-metal additives. Therefore, we propose that the IncHI2 plasmids were circulating in the surrounding environment, or were resident in the faecal flora of sows. These external factors are likely to be crucial in the acquisition of multiple drug resistance by new-born pigs. Therefore, it is imperative that MDR *E. coli* and the mobile elements they carry are tracked to determine their routes of entry into the gastrointestinal tracts of humans, companion and farmed animals and wildlife and into the food chain.

The CRL described in pTZ41_1P_HI2 may have originated by multiple homologous recombination events, however, defining exact genealogical events that may have formed the structure would be speculative. The presence of a Tn*6029-like* transposon within the hybrid Tn*1721*/Tn*21* transposon and the association with a *sul3*-integron in pTZ41_1P_HI2, highlights the wide distribution of Tn*6029* and the related transposon Tn*6026* in different plasmid backbones in *E. coli* sourced from diverse hosts [39,42]. Hybrid Tn*1721*/Tn*21* transposons have been reported in multiple bacterial hosts including *E. coli* and *Acinetobacter baumannii* [19,27,37,52,53] and are known to be important purveyors of ARGs on different plasmid backbones. Distinguishing features of the hybrid transposon included a Tn*21*/Tn*1721* hybrid resolution site, the Tn*1721*-specific transposition and tetracycline modules, and the Tn*21*-linked mercury resistance module.

*sul3*-integrons are widely distributed within commensal and pathogenic *Enterobacteriaceae* from humans [54], a range of food animals including cattle [55], swine [8] and poultry [56] and white stork [46]. The *sul3* gene was reported not only from specimens collected in animal farms, but also from surface sediment in lakes where Cu and Zn resistance genes [57] were present, and wetlands constructed to treat pig manure [58]. Notably, *sul3* has been detected in tap water, highlighting its importance as an emerging environmental pollutant [59]. The *sul3*- integrons previously reported in *E. coli* sourced from Australian pigs, typically exhibit IS*26*-mediated truncations in *mefB* of 111 bp and 260 bp [8,19]. In addition, *sul3*- integrons housing identical *mefB* deletions have been described overseas in pigs and humans [43,60], highlighting possible movement of plasmids and transposons that carry these *sul3* integrons. To our knowledge, the *sul3*- integron harbouring a full copy of *mefB* described in our study has not been previously reported in Australia.

Tellurium compounds are rarely present in the environment yet *ter* genes are found in different pathogenic bacterial species, suggesting an important but uncharacterised biological role for the gene [35,61]. The IncHI2 backbone structure is considered to be stable [20]. As such it is likely that the presence of different ARGs in these plasmids arose due to the acquisition of different mobile genetic elements [62,63,64].

The association of silver and copper resistance genes with Tn*7*-like transposons was reported in IncHI2:ST3 plasmids from *E. coli* and *Salmonella enterica* of porcine origin in Australia [19,23,65], and China [24] indicating that Tn*7*-like transposons play an important role in the spread of heavy-metal resistance genes. Heavy-metals, such as copper and silver are found in disinfectants, soil fertilisers and livestock feed and are recognised environmental pollutants. Additionally, they have been used as growth-promotion agents and for the prevention of post-weaning diarrheal disease in swine [66,67,68]. As such, it is common to identify bacterial populations in the gastrointestinal flora of swine and other intensively reared food-animal species with genetic mechanisms to survive and proliferate in their presence. Moreover, the ability of IncHI2 plasmids to conjugate at temperatures well below 37 °C may enhance their ability to acquire heavy-metal resistance genes from the environment [22,23]. Our analysis and other recent studies [19,24,25,30,69] suggest that the presence of Tn*7*-like transposons carrying *pco* and *sil* operons in IncHI2 plasmids reported in swine in Australia and China may have arisen due to co-selection pressures afforded by the use of these metals in feed additives. Additionally, arsenic compounds have been used as feed additives for swine production and were present in the environment [70].

Plasmids, such as pTZ41_1P_HI2, carry CRL that in part, were assembled by the action of IS*26*. Relentless antimicrobial selective pressures in intensive animal industries are likely playing important roles as drivers of plasmid fitness as these plasmids are commonly retained by commensal enterobacterial populations. IS*26*, via its ability to initiate DNA deletions, may play a role in altering plasmid fitness by eliminating genetic traits that impose a fitness impost [71]. Because the post-weaned piglet carrying the ST744 *E. coli* strain TZ41_1P received a ration comprising commercial solid feed on the day of sampling, we postulate the most likely opportunity for this animal to acquire TZ41_1P carrying pTZ41_1P_HI2 was via maternal flora or from flora from the surrounding environment at its source. Given the propensity of IncHI2 plasmids to capture genes encoding resistance to diverse ARGs, we recommend that further work is conducted to identify genetic signatures that can be used to track IncHI2 plasmids carrying diverse CRL both from commensal and pathogenic *E. coli* inhabiting the intestinal tracks of livestock, wastewater, wildlife, food and the environment more broadly.

pRDB9 and pTZ41_1P_HI2 share high sequence homology and a consistent group of ARGs despite their isolation from geographically distinct regions. Despite important differences underpinning antimicrobial usage policies in farming systems in Australia and Europe, these data suggest that plasmids can be mobilised over great distances facilitated in part by international travel and wildlife. Our analyses showed that components of Tn*1721* were identified in pRDB9, including tetracycline resistance genes *tetR, tetA*, but evidence of a hybrid transposon as seen in pTZ41_1P_HI2 was missing. The Tn*7-like sil/pco* transposon is present in pRDB9 but not associated with other transposons carrying ARGs and the *ars* operon was absent. Additionally, pRDB9 carried the *mcr-1* gene embedded in the composite transposon Tn*6330*. These and other observations highlight how highly similar plasmids adjust to the antimicrobial selection pressures imparted onto the host bacterium by altering the resistance gene cargo they carry [72,73].

IncHI2 plasmids are likely circulating in the gut of healthy pigs in Australia, as shown by the comparative analysis between pTZ41_1P_HI2 and pSDE-SvHI2. pSDE-SvHI2 and pTZ41_1P_HI2 were sourced 10 years apart from different *E. coli* lineages (ETEC and commensal *E. coli*, respectively), but their IncHI2 backbone and the overall gene content in the different modules that define the respective MDR region remained similar. IncHI2 plasmids are likely to be well-established in the gut microflora of Australian pig populations as they have been described in commensal and pathogenic *E. coli* [19,74] and in a dominant Australian lineage of *Salmonella enterica* serotype 1,4,5,12, [23] and are globally disseminated in food-producing animals [24]. Carriage of atypical class 1 *sul3* integrons, multiple transposons and heavy metal resistance operons are a feature of these plasmids.

Recently, carriage of ARGs and heavy-metal resistance genes embedded in genomic structures similar to pTZ41_1P_HI2, were reported in a IncHI2 ST3 Australian plasmid sequence of porcine origin [19]. The class 1 integron *sul3*-Δ*mefB*_111_ hosted by a Tn*7-like* transposon appeared to be a feature unique to Australian porcine *E. coli*. These plasmids carry genomic signatures within CRL that enable them to be tracked when transferred to other Inc-type plasmids or bacterial species. IncHI2 plasmids from different sources have conserved backbone structures and the resistance gene cargo that they carry reflects the antimicrobial selection pressure they were exposed to, brought about by the action of mobile genetic elements (MGE)s. More IncHI2 plasmids sequences are needed to better understand their role in AMR, heavy-metal resistance and virulence dissemination in healthy animals and in the environment on a global scale. Notable in this regard is the ability of IncHI2 plasmids to conjugate over a wide temperature range.

Strict antimicrobial stewardship practices and the ban on the import of live pigs in Australia represent important biosecurity measures that have limited the evolution and spread of clinically important antimicrobial resistance genes (ARGs) in Australian food animals. It is important to remain cognisant of the potential for human to animal transmission especially by veterinarians and farm workers that have recently returned from overseas from regions where carriage rates for ARGs encoding resistance to last-line clinically important antibiotics is a pressing problem.

## 5. Conclusions

We described for the first time an MDR IncHI2 ST4 plasmid in Australia. The characterisation of pTZ41_1P_HI2 contributes to the base knowledge of plasmids that circulate in intensive swine production systems in Australia. Our work underpins the importance of identifying and monitoring conjugative IncHI2 plasmids for their ability to capture and assemble a wide variety of ARGs and highlights their potential threat to human health. The widespread carriage of IS*26* on IncHI2 plasmids in Australia serves as a reminder of how readily these plasmids may acquire genes encoding resistance to clinically important antibiotics

Sound antimicrobial stewardship in Australian husbandry practices may have limited the acquisition of resistance to last-line agents. However, the high plasticity of IncHI2 plasmids and their capacity to adapt to diverse niches makes acquisition of clinically-important resistance genes by Australian IncHI2 plasmids a distinct future possibility.

## Figures and Tables

**Figure 1 microorganisms-08-00896-f001:**
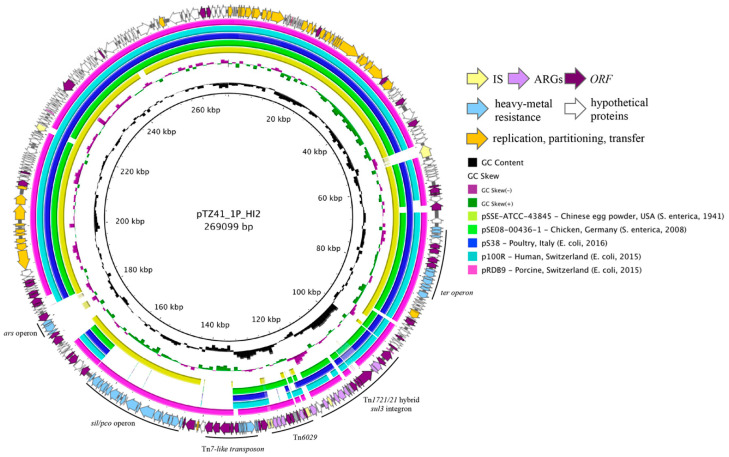
Circular map of pTZ41_1P_HI2 (outer ring with arrows) and pairwise BLASTn alignment of pTZ41_1P_HI2 with closely related plasmids represented as inner circles. Coloured arrows in the outer ring represent different gene families. A key of the coloured arrows representing different gene families is presented in the inset. The inner coloured circles representing different plasmids are also listed in the inset.

**Figure 2 microorganisms-08-00896-f002:**
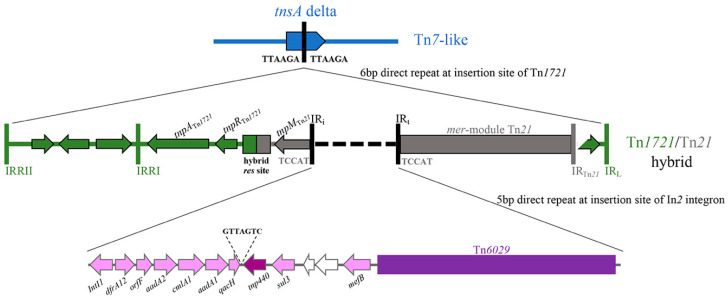
Diagrammatic representation of the resistance genes, class 1 integron, transposons and insertion elements that form the complex resistance locus on the plasmid.

**Figure 3 microorganisms-08-00896-f003:**
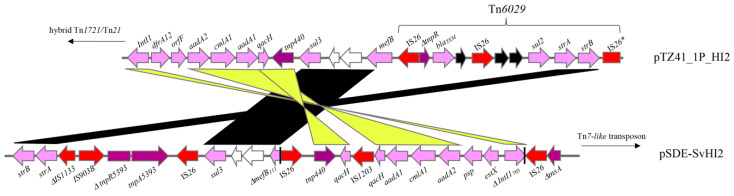
Pairwise alignment of the resistance locus in pTZ41_1P_HI2 and pSDE-SvHI2, highlighting regions of homology and differences.

**Figure 4 microorganisms-08-00896-f004:**
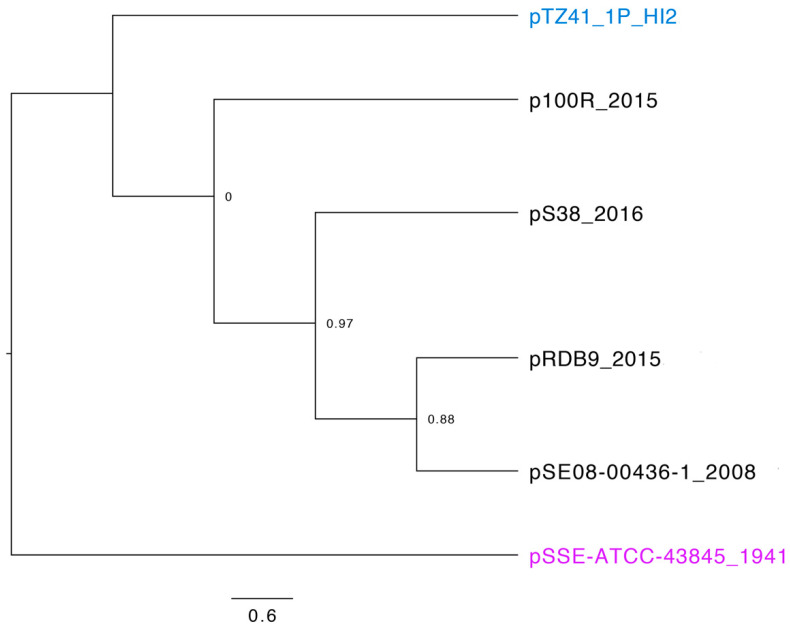
A maximum-likelihood phylogenetic tree of closely related IncHI2-ST4 plasmid sequences downloaded from GenBank. The tree was rooted using a sequence of the oldest known plasmid pSSE-ATCC-43845 belonging to the IncHI2-ST4 lineage in 1941. Node values represent confidence scores. The scale represents substitutions per site.

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
