# Peer review of "Genomic Characterisation of a Multiple Drug Resistant IncHI2 ST4 Plasmid in Escherichia coli ST744 in Australia"

_microorganisms, 2020, doi:10.3390/microorganisms8060896_

Round 1
Reviewer 1 Report
The study by Zingali et al. entitled “Genomic characterisation of a multiple drug resistant 3 IncHI2 ST4 plasmid in Escherichia coli ST744 in 4 Australia” describes the first complete Australian sequence of a multiple drug resistance 21 IncHI2-ST4 plasmid, pTZ41_1P from a commensal E. coli from a healthy piglet.
The topic is very important, emerging of MDR plasmids are a concern. The manuscript is very well written. However, the study would be stronger if authors provided some phenotypic data, like the transfer of the plasmid, it prevalence, etc…
Lines 80-81: “The pTZ41_1P_HI2 was derived from a commensal E. coli, sourced from the feces of a healthy four-week old piglet in Australia in 2017” Authors should provide more details on the isolate and the isolation conditions. Do authors know whether the strain contains only one plasmid, was a plasmid profile done on the isolates? What is the prevalence of the MDR bacteria? Was it found in only one animal or in multiple animals etc….and this section should be moved to M&M.
Lines 92-93: Not clear. Was the whole E. coli DNA sequenced or just the plasmid? was the plasmid purified? Please provide more clarification to this section.
Line 116. “IncHI2 ST4 plasmids sequences” are there more than one plasmid???
Lines 127-133: This section should be in M&M
Lines 148-149: For clarity, list all AGRs genes associated with the plasmids and discuss them as done. The way it is written they are lost in the paragraphs
Line 236: “whichcarries” leave a space between which and carries
Lines 287-288: Authors have not tested the transfer of the plasmids in different conditions.
Line 374: references : there are too many, just provide the most important ones. Also bacterial names are not italicized. Please revise,
Author Response
Response to Reviewer 1 Comments
Point 1: The topic is very important, emerging of MDR plasmids are a concern. The manuscript is very well written. However, the study would be stronger if authors provided some phenotypic data, like the transfer of the plasmid, it prevalence, etc…
Response: The project had a strong bio-informatic focus and hence wet lab experiments were not included. However, we would like to draw the reviewer’s attention to published (cited) literature on IncHI2 plasmids, which iteratively prove the ability of IncHI2 group of broad host range plasmids to conjugate into various members of Enterobacterales. As a part of this analysis though, we have analysed genes which form the plasmid backbone and support lateral transfer and have clearly mentioned the homology of such genes with well characterised IncHI2 plasmid/s.
There is a paucity of reports on completely closed and characterised IncHI2-ST4 plasmids from porcine sources, both locally and globally, and this manuscript bridges that gap. We have highlighted the issue in the discussion section. More effort needs to be channelled in closing multi-resistant conjugative plasmids for surveillance-based research to generate data-driven evidence to map their relevance in the dissemination of MDR within bacterial populations.
Point 2: Lines 80-81: “The pTZ41_1P_HI2 was derived from a commensal E. coli, sourced from the feces of a healthy four-week old piglet in Australia in 2017” Authors should provide more details on the isolate and the isolation conditions. Do authors know whether the strain contains only one plasmid, was a plasmid profile done on the isolates? What is the prevalence of the MDR bacteria? Was it found in only one animal or in multiple animals etc….and this section should be moved to M&M.
RESPONSE: As this manuscript is focussed on the IncHI2-ST4 plasmid we have chosen to keep it this focussed on it. However, details of the strain and genotypic features including plasmid profile of the isolate (TZ41_1P) has recently been published in Microorganisms (Ziangali et al, doi:10.3390/microorganisms8060843.) and is referenced as item number 10 in the bibliography list.
We have additionally moved first subsection of the results to the methods section, as suggested in point 5 (below) in the revised version of the manuscript.
Point 3: Lines 92-93: Not clear. Was the whole E. coli DNA sequenced or just the plasmid? was the plasmid purified? Please provide more clarification to this section.
RESPONSE: We have clarified this in the methods section in lines 97-100 of the revised manuscript. It now reads:
“The complete sequence of pTZ41_1P_HI2 was assembled using the Unicycler v0.4.6 [31] hybrid assembler with raw reads from Illumina and Nanopore sequencing runs generating a completely closed plasmid sequence. For illumina short read sequencing, genomic DNA from Escherichia coli TZ41_1P was isolated using a DNeasy Blood and Tissue Kit (Qiagen) following the manufacturer’s protocol.”
Point 4: Line 116. “IncHI2 ST4 plasmids sequences” are there more than one plasmid???
RESPONSE: We refer to plasmids downloaded from GenBank. We have now amended the sentence to clarify it. It appears in line 122 in the revised version of the manuscript.
Point 5: Lines 127-133: This section should be in M&M
RESPONSE: We have shifted the section. It now appears as subsection 1 of Material and Methods in lines 91-95, as below:
2.1. Plasmid pTZ41_1P_HI2:
pTZ41_1P_HI2 was identified from a commensal phylogroup A E. coli (strain TZ41_1P) belonging to ST744 and serotype O89:H25. The strain was sourced from faecal material of a four-week old piglet, and part of a collection of commensal E. coli that carry the class 1 integrase intI1+. The cohort of pigs were sourced from an Australian commercial breeding operation in January 2017. Pigs were not previously treated with antimicrobials and weaned on the day before the initial faecal sampling.
Point 6: Lines 148-149: For clarity, list all AGRs genes associated with the plasmids and discuss them as done. The way it is written they are lost in the paragraphs
RESPONSE: We have now listed the resistance genes in the CRL. The amendment appears in lines 147-149 in the revised version of the manuscript and reads:
The CRL comprised genes capable of conferring resistance to trimethoprim (dfrA12), chloramphenicol (cmlA1), ampicillin (blaTEM-1) aminoglycoside antibiotics streptomycin (aadA1, strA and strB) and spectinomycin (aadA1) and sulfonamides (sul2, sul3).
Point 7: Line 236: “whichcarries” leave a space between which and carries
RESPONSE: We have included the space. Please see line 244 in the revised version of the manuscript.
Point 8: Lines 287-288: Authors have not tested the transfer of the plasmids in different conditions.
RESPONSE: We are guessing this comment is based on lines 309-311 in the revised manuscript. No, conjugation experiments were not conducted. This was a general discussion point based on published literature without any claim of tests for plasmid transfer.
Point 9: Line 374: references : there are too many, just provide the most important ones. Also bacterial names are not italicized. Please revise,
RESPONSE: We have reduced the reference list to 74 and have italicized bacteria names in the revised version.
Reviewer 2 Report
congratulations for this thorough work.
I might only suggest rewriting lines 81 to 84 of the manuscript to make it easy to understand the sentence.
Author Response
We have ammended the sections suggested, which appears in line 80-84 in the revised version and reads:
"Here we report the complete sequence of pTZ41_1P_HI2, the first MDR IncHI2 ST4 plasmid characterised from a porcine commensal in Australia. pTZ41_1P_HI2 was isolated from an E. coli collected from the faeces of a healthy four-week old piglet in 2017. We present the phylogeny of pTZ41_1P_HI2 and other IncHI2 ST4 plasmid sequences available in public repositories We also present a detailed comparison of the drug resistance locus in pTZ41_1P with that found in closely related plasmid pSDE-SvHI2, an IncHI2:ST3 plasmid isolated from a pathogenic porcine ETEC/ExPEC in Australia in 2007 [19,30]. "